# How to train your MAML

**Antreas Antoniou**
University of Edinburgh
{a.antoniou}@sms.ed.ac.uk

**Harrison Edwards**
OpenAI, University of Edinburgh
{h.l.edwards}@sms.ed.ac.uk

**Amos Storkey**
University of Edinburgh
{a.storkey}@ed.ac.uk

## ABSTRACT

The field of few-shot learning has recently seen substantial advancements. Most of these advancements came from casting few-shot learning as a meta-learning problem. *Model Agnostic Meta Learning* or MAML is currently one of the best approaches for few-shot learning via meta-learning. MAML is simple, elegant and very powerful, however, it has a variety of issues, such as being very sensitive to neural network architectures, often leading to instability during training, requiring arduous hyperparameter searches to stabilize training and achieve high generalization and being very computationally expensive at both training and inference times. In this paper, we propose various modifications to MAML that not only stabilize the system, but also substantially improve the generalization performance, convergence speed and computational overhead of MAML, which we call *MAML++*.

## 1 INTRODUCTION

The human capacity to learn new concepts using only a handful of samples is immense. In stark contrast, modern deep neural networks need, at a minimum, thousands of samples before they begin to learn representations that can generalize well to unseen data-points (Krizhevsky et al., 2012; Huang et al., 2017), and mostly fail when the data available is scarce. The fact that standard deep neural networks fail in the small data regime can provide hints about some of their potential shortcomings. Solving those shortcomings has the potential to open the door to understanding intelligence and advancing Artificial Intelligence. Few-shot learning encapsulates a family of methods that can learn new concepts with only a handful of data-points (usually 1-5 samples per concept). This possibility is attractive for a number of reasons. First, few-shot learning would reduce the need for data collection and labelling, thus reducing the time and resources needed to build robust machine learning models. Second, it would potentially reduce training and fine-tuning times for adapting systems to newly acquired data. Third, in many real-world problems there are only a few samples available per class and the collection of additional data is either remarkably time-consuming and costly or altogether impossible, thus necessitating the need to learn from the available few samples.

The nature of few-shot learning makes it a very hard problem if no prior knowledge exists about the task at hand. For a model to be able to learn a robust model from a few samples, knowledge transfer (see e.g. Caruana, 1995) from other similar tasks is key. However, manual knowledge transfer from one task to another for the purpose of fine-tuning on a new task can be a time consuming and ultimately inefficient process. *Meta-learning* (Schmidhuber, 1987; Vilalta & Drissi, 2002), or *learning to learn* (Thrun & Pratt, 1998), can instead be used to automatically learn across-task knowledge usually referred to as *across-task* (or sometimes slow) knowledge such that our model can, at inference time, quickly acquire *task-specific* (or fast) knowledge from new tasks using only a few samples. Meta-learning can be broadly defined as a class of machine learning models that become more proficient at learning with more experience, thus learning *how* to learn. More specifically meta-learning involves learning at two levels. At the *task-level*, where the *base-model* is required to acquire task-specific (fast) knowledge rapidly, and at the *meta-level*, where the *meta-model* is required to slowly learn *across-task* (slow) knowledge. Recent work in meta-learning has produced

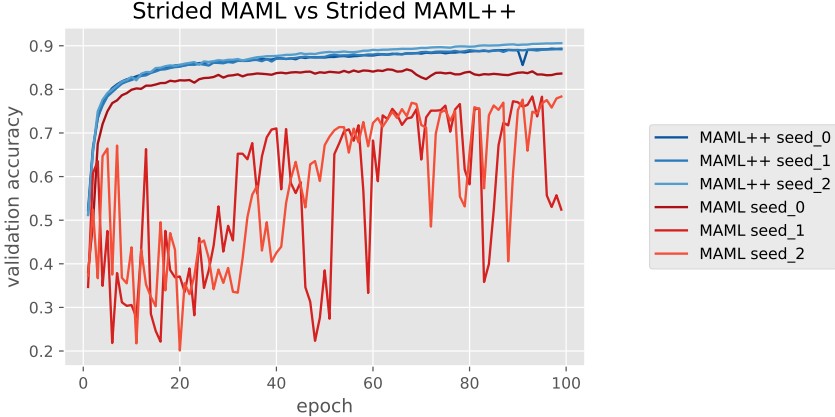

Figure 1: Stabilizing MAML: This figure illustrates 3 seeds of the original strided MAML vs strided MAML++. One can see that 2 out of 3 seeds with the original strided MAML seem to become unstable and erratic, whereas all 3 of the strided MAML++ models seem to consistently converge very fast, to much higher generalization accuracy without any stability issues.

state of the art results in a variety of settings (Wang et al., 2016; Ba et al., 2016; Zoph & Le, 2016; Andrychowicz et al., 2016; Vinyals et al., 2016; Li & Malik, 2016; Li et al., 2017; Antoniou et al., 2017; Brock et al., 2017; Munkhdalai & Yu, 2017). The application of meta-learning in the few-shot learning setting has enabled the overwhelming majority of the current state of the art few-shot learning methods (Vinyals et al., 2016; Ravi & Larochelle, 2016; Edwards & Storkey, 2016; Finn et al., 2017; Li et al., 2017; Munkhdalai & Yu, 2017). One such method, known for its simplicity and state of the art performance, is *Model Agnostic Meta-Learning* (*MAML*) (Finn et al., 2017). In MAML, the authors propose learning an initialization for a base-model such that after applying a very small number of gradient steps with respect to a training set on the base-model, the adapted model can achieve strong generalization performance on a validation set (the validation set consists of new samples from the same classes as the training set). Relating back to the definitions of meta-model and base-model, in MAML the meta-model is effectively the initialization parameters. These parameters are used to initialize the base-model, which is then used for task-specific learning on a support set, which is then evaluated on a target set. MAML is a simple yet elegant meta-learning framework that has achieved state of the art results in a number of settings. However, MAML suffers from a variety of problems which: 1) cause instability during training, 2) restrict the model's generalization performance, 3) reduce the framework's flexibility, 4) increase the system's computational overhead and 5) require that the model goes through a costly (in terms of time and computation needed) hyperparameter tuning before it can work robustly on a new task.

In this paper we propose MAML++, an improved variant of the MAML framework that offers the flexibility of MAML along with many improvements, such as robust and stable training, automatic learning of the inner loop hyperparameters, greatly improved computational efficiency both during inference and training and significantly improved generalization performance. MAML++ is evaluated in the few-shot learning setting where the system is able to set a new state of the art across all established few-shot learning tasks on both Omniglot and Mini-Imagenet, performing as well as or better than all established meta learning methods on both tasks.

## 2 RELATED WORK

The *set-to-set* few-shot learning setting (Vinyals et al., 2016), neatly casts few-shot learning as a meta-learning problem. In set-to-set few-shot learning we have a number of tasks, each task is composed by a *support set* which is used for task-level learning, and a *target set* which is used for evaluating the base-model on a certain task after it has acquired task-specific (or fast) knowledge. Furthermore, all available tasks are split into 3 sets, the *meta-training* set, the *meta-validation* set and the *meta-test* set, used for training, validating and testing our meta-learning model respectively.

Once meta-learning was shown to be an effective framework for few-shot learning and the set to set approach was introduced, further developments in few-shot learning were made in quick succession. One contribution was *Matching Networks* (Vinyals et al., 2016). Matching networks achieve few-shot learning by learning to *match* target set items to support set items. More specifically, a matching network learns to match the target set items to the support set items using cosine distance and a fully differentiable embedding function. First, the support set embedding function $g$, parameterized as a deep neural network, embeds the support set items into embedding vectors, then the target set embedding function $f$ embeds the target set items. Once all data-item embeddings are available, cosine distance is computed for all target set embeddings when compared to all support set embeddings. As a result, for each target set item, a vector of cosine distances with respect to all support set items will be generated (with each distance's column tied to the respective support set class). Then, the softmax function is applied on the generated distance vectors, to convert them into probability distributions over the support set classes.

Another notable advancement was the gradient-conditional *meta-learner* LSTM (Ravi & Larochelle, 2016) that learns how to update a *base-learner* model. At inference time, the meta-learner model applies a single update on the base-learner given gradients with respect to the support set. The fully updated base-model then computes predictions on the target set. The target set predictions are then used to compute a task loss. Furthermore they jointly learn the meta-learner's parameters as well as the base-learners initializations such that after a small number of steps it can do very well on a given task. The authors ran experiments on Mini-Imagenet where they exceed the performance of Matching Networks.

In *Model Agnostic meta-learning* (MAML) (Finn et al., 2017) the authors proposed increasing the gradient update steps on the base-model and replacing the meta-learner LSTM with Batch Stochastic Gradient Descent (Krizhevsky et al., 2012), which as a result speeds up the process of learning and interestingly improves generalization performance and achieves state of the art performance in both Omniglot and Mini-Imagenet.

In *Meta-SGD* (Li et al., 2017) the authors proposed learning a static learning rate and an update direction for each parameter in the base-model, in addition to learning the initialization parameters of the base-model. Meta-SGD showcases significantly improved generalization performance (when compared to MAML) across all few-shot learning tasks, whilst only requiring a single inner loop update step. However this practice effectively doubles the model parameters and computational overheads of the system.

## 3   MODEL AGNOSTIC META LEARNING

*Model Agnostic Meta-Learning* (MAML) (Finn et al., 2017) is a meta-learning framework for few-shot learning. MAML is elegantly simple yet can produce state of the art results in few-shot regression/classification and reinforcement learning problems. In a sentence, MAML learns good initialization parameters for a network, such that after a few steps of standard training on a few-shot dataset, the network will perform well on that few shot task.

More formally, we define the *base model* to be a neural network $f_\theta$ with meta-parameters $\theta$. We want to learn an initial $\theta = \theta_0$ that, after a small number $N$ of gradient update steps on data from a support set $S_b$ to obtain $\theta_N$, the network performs well on that task's target set $T_b$. Here $b$ is the index of a particular support set task in a batch of support set tasks. This set of $N$ updates steps is called the *inner-loop update process*.

The updated base-network parameters after $i$ steps on data from the support task $S_b$ can be expressed as:

$$\theta_i^b = \theta_{i-1}^b - \alpha \nabla_\theta \mathcal{L}_{S_b}(f_{\theta_{i-1}^b}),\tag{1}$$

where $\alpha$ is the learning rate, $\theta_i^b$ are the base-network weights after $i$ steps towards task $b$, $\mathcal{L}_{S_b}(f_{\theta_{(i-1)}})$ is the loss on the support set of task $b$ after $(i-1)$ (i.e. the previous step) update steps. Assuming that our *task batch size* is $B$ we can define a *meta-objective*, which can be expressed as:

$$\mathcal{L}_{meta}(\theta_0) = \sum_{b=1}^{B} \mathcal{L}_{T_b}(f_{\theta_N^b(\theta_0)})\tag{2}$$

where we have explicitly denoted the dependence of $\theta_N^b$ on $\theta_0$, given by unrolling (1). The objective (2) measures the quality of an initialization $\theta_0$ in terms of the total loss of using that initialization across all tasks. This meta objective is now minimized to optimize the initial parameter value $\theta_0$. It is this initial $\theta_0$ that contains the across-task knowledge. The optimization of this meta-objective is called the *outer-loop update process*.

The resulting update for the meta-parameters $\theta_0$ can be expressed as:

$$\theta_0 = \theta_0 - \beta \nabla_\theta \sum_{b=1}^{B} \mathcal{L}_{T_b}(f_{\theta_N^b(\theta_0)}) \tag{3}$$

where $\beta$ is a learning rate and $\mathcal{L}_{T_b}$ denotes the loss on the target set for task $b$.

In this paper we use the cross-entropy (De Boer et al., 2005; Rubinstein, 1999) loss throughout.

## 3.1 MODEL AGNOSTIC META-LEARNING PROBLEMS

The simplicity, elegance and high performance of MAML make it a very powerful framework for meta-learning. However, MAML has also many issues that make it problematic to use.

**Training Instability:** Depending on the neural network architecture and the overall hyperparameter setup, MAML can be very unstable during training as illustrated in Figure 1. Optimizing the outer loop involved backpropagating derivatives through an unfolded inner loop consisting of the same network multiple times. This alone could be cause for gradient issues. However, the gradient issues are further compounded by the model architecture, which is a standard 4-layer convolutional network without skip-connections. The lack of any skip-connections means that every gradient must be passed through each convolutional layer many times; effectively the gradients will be multiplied by the same sets of parameters multiple times. After multiple back-propagation passes, the large depth structure of the unfolded network and lack of skip connections can cause gradient explosions and diminishing gradient problems respectively.

**Second Order Derivative Cost:** Optimization through gradient update steps requires the computation of second order gradients which are very expensive to compute. The authors of MAML proposed using first-order approximations to speed up the process by a factor of three, however using these approximations can have a negative impact on the final generalization error. Further attempts at using first order methods have been attempted in Reptile (Nichol et al., 2018) where the authors apply standard SGD on a base-model and then take a step from their initialization parameters towards the parameters of the base-model after $N$ steps. The results of Reptile vary, in some cases exceeding MAML, and in others producing results inferior to MAML. Approaches to reduce computation time while not sacrificing generalization performance have yet to be proposed.

**Absence of Batch Normalization Statistic Accumulation:** A further issue that affects the generalization performance is the way that batch normalization is used in the experiments in the original MAML paper. Instead of accumulating running statistics, the statistics of the current batch were used for batch normalization. This results in batch normalization being less effective, since the biases learned have to accommodate for a variety of different means and standard deviations instead of a single mean and standard deviation. On the other hand, if batch normalization uses accumulated running statistics it will eventually converge to some global mean and standard deviation. This leaves only a single mean and standard deviation to learn biases for. Using running statistics instead of batch statistics, can greatly increase convergence speed, stability and generalization performance as the normalized features will result in smoother optimization landscape (Santurkar et al., 2018).

**Shared (across step) Batch Normalization Bias:** An additional problem with batch normalization in MAML stems from the fact that batch normalization biases are not updated in the inner-loop; instead the same biases are used throughout all iterations of base-models. Doing this implicitly assumes that all base-models are the same throughout the inner loop updates and hence have the same distribution of features passing through them. This is a false assumption to make, since, with each inner loop update, a new base-model is instantiated that is different enough from the previous one to be considered a new model from a bias estimation point of view. Thus learning a single set of biases for all iterations of the base-model can restrict performance.

**Shared Inner Loop (across step and across parameter) Learning Rate:** One issue that affects both generalization and convergence speed (in terms of training iterations) is the issue of using a

shared learning rate for all parameters and all update-steps. Doing so introduces two major problems. Having a fixed learning rate requires doing multiple hyperparameter searches to find the correct learning rate for a specific dataset; this process can be very computationally costly, depending on how search is done.

The authors in (Li et al., 2017) propose to learn a learning rate and update direction for each parameter of the network. Doing so solves the issue of manually having to search for the right learning rate, and also allows individual parameters to have smaller or larger learning rates. However this approach brings its own problems. Learning a learning rate for each network parameter means increased computational effort and increased memory usage since the network contains between 40K and 50K parameters depending on the dimensionality of the data-points.

**Fixed Outer Loop Learning Rate:** In MAML the authors use Adam with a fixed learning rate to optimize the meta-objective. Annealing the learning rate using either step or cosine functions has proven crucial to achieving state of the art generalization performance in a multitude of settings (Loshchilov & Hutter, 2016; He et al., 2016; Larsson et al., 2016; Huang et al., 2017). Thus, we theorize that using a static learning rate reduces MAML's generalization performance and might also be a reason for slower optimization. Furthermore, having a fixed learning rate might mean that one has to spend more (computational) time tuning the learning rate.

## 4 STABLE, AUTOMATED AND IMPROVED MAML

In this section we propose methods for solving the issues with the MAML framework, described in Section 3.1. Each solution has a reference identical to the reference of the issue it is attempting to solve.

**Gradient Instability → Multi-Step Loss Optimization (MSL)**: MAML works by minimizing the target set loss computed by the base-network after it has completed **all** of its inner-loop updates towards a support set task. Instead we propose minimizing the target set loss computed by the base-network after **every** step towards a support set task. More specifically, we propose that the loss minimized is a weighted sum of the target set losses after every support set loss update. More formally:

$$\theta = \theta - \beta \nabla_\theta \sum_{b=1}^{B} \sum_{i=0}^{N} v_i \mathcal{L}_{T_b}(f_{\theta_i^b}) \tag{4}$$

Where $\beta$ is a learning rate, $\mathcal{L}_{T_b}(f_{\theta_i^b})$ denotes the target set loss of task $b$ when using the base-network weights after $i$ steps towards minimizing the support set task and $v_i$ denotes the importance weight of the target set loss at step $i$, which is used to compute the weighted sum.

By using the *multi-step* loss proposed above we improve gradient propagation, since now the base-network weights at every step receive gradients both directly (for the current step loss) and indirectly (from losses coming from subsequent steps). With the original methodology described in Section 3 the base-network weights at every step except the last one were optimized implicitly as a result of backpropagation, which caused many of the instability issues MAML had. However using the multi-step loss alleviates this issue as illustrated in Figure 1. Furthermore, we employ an annealed weighting for the per step losses. Initially all losses have equal contributions towards the loss, but as iterations increase, we decrease the contributions from earlier steps and slowly increase the contribution of later steps. This is done to ensure that as training progresses the final step loss receives more attention from the optimizer thus ensuring it reaches the lowest possible loss. If the annealing is not used, we found that the final loss might be higher than with the original formulation.

**Second Order Derivative Cost → Derivative-Order Annealing (DA):** One way of making MAML more computationally efficient is reducing the number of inner-loop updates needed, which can be achieved with some of the methods described in subsequent sections of this report. However, in this paragraph, we propose a method that reduces the per-step computational overhead directly. The authors of MAML proposed the usage of first-order approximations of the gradient derivatives. However they applied the first-order approximation throughout the whole of the training phase. Instead, we propose to anneal the derivative-order as training progresses. More specifically, we propose to use first-order gradients for the first 50 epochs of the training phase, and to then switch to second-order gradients for the remainder of the training phase. We empirically demonstrate that

doing so greatly speeds up the first 50 epochs, while allowing the second-order training needed to achieve the strong generalization performance the second-order gradients provide to the model. An additional interesting observation is that derivative-order annealing experiments showed no incidents of exploding or diminishing gradients, contrary to second-order only experiments which were more unstable. Using first-order before starting to use second-order derivatives can be used as a strong *pretraining* method that learns parameters less likely to produce gradient explosion/diminishment issues.

**Absence of Batch Normalization Statistic Accumulation → Per-Step Batch Normalization Running Statistics (BNRS):** In the original implementation of MAML Finn et al. (2017) the authors used only the current batch statistics as the batch normalization statistics. This, we argue, caused a variety of undesirable effects described in Section 3.1. To alleviate the issues we propose using running batch statistics for batch normalization. A naive implementation of batch normalization in the context of MAML would require sharing running batch statistics across all update steps of the inner-loop fast-knowledge acquisition process. However doing so would cause the undesirable consequence that the statistics stored be shared across all inner loop updates of the network. This would cause optimization issues and potentially slow down or altogether halt optimization, due to the increasing complexity of learning parameters that can work across various updates of the network parameters. A better alternative would be to collect statistics in a per-step regime. To collect running statistics per-step, one needs to instantiate $N$ (where $N$ is the total number of inner-loop update steps) sets of running mean and running standard deviation for each batch normalization layer in the network and update the running statistics respectively with the steps being taken during the optimization. The per-step batch normalization methodology should speed up optimization of MAML whilst potentially improving generalization performance.

**Shared (across step) Batch Normalization Bias → Per-Step Batch Normalization Weights and Biases (BNWB):** In the MAML paper the authors trained their model to learn a **single** set of biases for each layer. Doing so assumes that the distributions of features passing through the network are similar. However, this is a false assumption since the base-model is updated for a number of times, thus making the feature distributions increasingly dissimilar from each other. To fix this problem we propose learning a set of biases **per-step** within the inner-loop update process. Doing so, means that batch normalization will learn biases specific to the feature distributions seen at each set, which should increase convergence speed, stability and generalization performance.

**Shared Inner Loop Learning Rate (across step and across parameter) → Learning Per-Layer Per-Step Learning Rates and Gradient Directions (LSLR):** Previous work in Li et al. (2017) demonstrated that learning a learning rate and gradient direction for each parameter in the base-network improved the generalization performance of the system. However, that had the consequence of increased number of parameters and increased computational overhead. So instead, we propose, learning a learning rate and direction for each layer in the network as well as learning different learning rates for each adaptation of the base-network as it takes steps. Learning a learning rate and direction for each layer instead for each parameter should reduce memory and computation needed whilst providing additional flexibility in the update steps. Furthermore, for each learning rate learned, there will be $N$ instances of that learning rate, one for each step to be taken. By doing this, the parameters are free to learn to decrease the learning rates at each step which may help alleviate overfitting.

**Fixed Outer Loop Learning Rate → Cosine Annealing of Meta-Optimizer Learning Rate (CA):** In MAML the authors use a static learning rate for the optimizer of the meta-model. Annealing the learning rate, either by using step-functions (He et al., 2016) or cosine functions (Loshchilov & Hutter, 2016) has proved vital in learning models with higher generalization power. The cosine annealing scheduling has been especially effective in producing state of the art results whilst removing the need for any hyper-parameter searching on the learning rate space. Thus, we propose applying the cosine annealing scheduling on the meta-model's optimizer (i.e. the *meta-optimizer*). Annealing the learning rate allows the model to fit the training set more effectively and as a result might produce higher generalization performance.

### 4.1 DATASETS

The datasets used to evaluate our methods were the Omniglot (Lake et al., 2015) and Mini-Imagenet (Vinyals et al., 2016; Ravi & Larochelle, 2016) datasets. Each dataset is split into 3 sets, a training, validation and test set. The Omniglot dataset is composed of 1623 characters classes from various alphabets. There exist 20 instances of each class in the dataset. For Omniglot we shuffle all character classes and randomly select 1150 for the training set and from the remaining classes we use 50 for validation and 423 for testing. In most few-shot learning papers the first 1200 classes are used for training and the remaining for testing. However, having a small validation set to choose the best model is crucial, so we choose to use a small set of 50 classes as validation set. For each class we use all available 20 samples in the sets. Furthermore for the Omniglot dataset, data augmentation is used on the images in the form of rotations of 90 degree increments. Class samples that are rotated are considered new classes, e.g. a 180 degree rotated character $C$ is considered a different class from a non rotated $C$, thus effectively having 1623 x 4 classes in total. However the rotated classes are generated dynamically after the character classes have been split into the sets such that rotated samples from a class reside in the same set (i.e. the training, validation or test set). The Mini-Imagenet dataset was proposed in Ravi & Larochelle (2016), it consists of 600 instances of 100 classes from the ImageNet dataset, scaled down to 84x84. We use the split proposed in Ravi & Larochelle (2016), which consists of 64 classes for training, 12 classes for validation and 24 classes for testing.

### 4.2 EXPERIMENTS

To evaluate our methods we adopted a hierarchical hyperparameter search methodology. First we began with the baseline MAML experiments, which were ran on the 5/20-way and 1/5-shot settings on the Omniglot dataset and the 5-way 1/5-shot setting on the Mini-Imagenet dataset. Then we added each one of our 6 methodologies on top of the default MAML and ran experiments for each one separately. Once this stage was completed we combined the approaches that showed improvements in either generalization performance or convergence speed (both in terms of number of epochs and clock-time) and ran a final experiment to establish any potential gains from the combination of the techniques.

An experiment consisted of training for 150 epochs, each epoch consisting of 500 iterations. At the end of each epoch, we evaluated the performance of the model on the validation set. Upon completion of all epochs, an ensemble of the top 3 performing per-epoch-models on the validation set were applied on the test set, thus producing the final test performance of the model. An evaluation ran consisted of inference on 600 unique tasks. A distinction between the training and evaluation tasks, was that the training tasks were generated dynamically continually without repeating previously sampled tasks, whilst the 600 evaluation tasks generated were identical across epochs. Thus ensuring that the comparison between models was fair, from an evaluation set viewpoint. Every experiment was repeated for 3 independent runs.

The models were trained using the Adam optimizer with a learning rate of 0.001, $\beta_1 = 0.9$ and $\beta_2 = 0.99$. Furthermore, all Omniglot experiments used a task batch size of 16, whereas for the Mini-Imagenet experiments we used a task batch size of 4 and 2 for the 5-way 1-shot and 5-way 5-shot experiments respectively.

### 4.3 RESULTS

Our proposed methodologies are empirically shown to improve the original MAML framework. In Table 1 one can see how our proposed approach performs on Omniglot. Each proposed methodology can individually outperform MAML, however, the most notable improvements come from the learned per-step per-layer learning rates and the per-step batch normalization methodology. In the 5-way 1-shot tasks it achieves 99.47% and in the 20-way Omniglot tasks MAML++ achieves 97.76% and 99.33% in the 1-shot and 5-shot tasks respectively. MAML++ also showcases improved convergence speed in terms of training iterations required to reach the best validation performance. Furthermore, the multi-step loss optimization technique substantially improves the training stability of the model as illustrated in Figure 1. In Table 1 we also include the results of our own implementation of MAML, which reproduces all results except the 20-way 1-shot Omniglot case. Difficulty in replicating the specific result has also been noted before in Jamal et al. (2018). We base our

Table 1: MAML++ Omniglot 20-way Few-Shot Results: Our reproduction of MAML appears to be replicating all the results except the 20-way 1-shot results. Other authors have come across this problem as well Jamal et al. (2018). We report our own base-lines to provide better relative intuition on how each method impacted the test accuracy of the model. We showcase how our proposed improvements individually improve on the MAML performance. Our method improves on the existing state of the art.

| Omniglot 20-way Few-Shot Classification | | |
|---|---|---|
| | **Accuracy** | |
| **Approach** | **1-shot** | **5-shot** |
| Siamese Nets | 88.2% | 97.0% |
| Matching Nets | 93.8% | 98.5% |
| Neural Statistician | 93.2% | 98.1% |
| Memory Mod. | 95.0% | 98.6% |
| Meta-SGD | 95.93±0.38% | 98.97±0.19% |
| Meta-Networks | 97.00% | − |
| MAML (original) | 95.8±0.3% | 98.9±0.2% |
| MAML (local replication) | 91.27±1.07% | 98.78% |
| MAML++ | **97.65±0.05%** | **99.33±0.03%** |
| MAML + MSL | 91.53±0.69% | - |
| MAML + LSLR | 95.77±0.38% | - |
| MAML + BNWB + BNRS | 95.35±0.23% | - |
| MAML + CA | 93.03±0.44% | - |
| MAML + DA | 92.3±0.55% | - |

Table 2: MAML++ Mini-Imagenet Results. *MAML++* indicates MAML + all the proposed fixes. Our reproduction of MAML appears to be replicating all the results of the original. Our approach sets a new state of the art across all tasks. It is also worth noting, that our approach, with only 1 inner loop step can already exceed all other methods. Additional steps allow for even better performance.

| Mini-Imagenet 5-way Few-Shot Classification | | | |
|---|---|---|---|
| | **Inner Steps** | **Accuracy** | |
| **Mini-Imagenet** | | **1-shot** | **5-shot** |
| Matching Nets | - | 43.56% | 55.31% |
| Meta-SGD | 1 | 50.47±1.87% | 64.03±0.94% |
| Meta-Networks | - | 49.21% | - |
| MAML (original paper) | 5 | 48.70±1.84% | 63.11±0.92% |
| MAML (local reproduction) | 5 | 48.25±0.62% | 64.39±0.31% |
| MAML++ | 1 | 51.05±0.31% | - |
| MAML++ | 2 | 51.49±0.25% | - |
| MAML++ | 3 | 51.11±0.11% | - |
| MAML++ | 4 | 51.65±0.34% | - |
| MAML++ | 5 | **52.15±0.26%** | **68.32±0.44%** |

conclusions on the relative performance between our own MAML implementation and the proposed methodologies.

Table 2 showcases MAML++on Mini-Imagenet tasks, where MAML++ sets a new state of the art in both the 5-way 1-shot and 5-shot cases where the method achieves 52.15% and 68.32% respectively. More notably, MAML++ can achieve very strong 1-shot results of 51.05% with only a single inner loop step required. Not only is MAML++ cheaper due to the usage of derivative order annealing, but also because of the much reduced inner loop steps. Another notable observation is that MAML++converges to its best generalization performance much faster (in terms of iterations required) when compared to MAML as shown in Figure 1.

## 5 Conclusion

In this paper we delve deep into what makes or breaks the MAML framework and propose multiple ways to reduce the inner loop hyperparameter sensitivity, improve the generalization error, stabilize and speed up MAML. The resulting approach, called MAML++sets a new state of the art across all few-shot tasks, across Omniglot and Mini-Imagenet. The results of the approach indicate that learning per-step learning rates, batch normalization parameters and optimizing on per-step target losses appears to be key for fast, highly automatic and strongly generalizable few-shot learning.

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

## A  ADDITIONAL RESULTS

Table 3: Mini-Imagenet Training Iteration Timing table. In this table, one can see per training iteration wall-clock timings for MAML vs MAML++. We provide timings for variants of the model spanning 1 to 5 inner loop steps. It can be observed that MAML++ needs less time per training iteration, even though it needs more parameters and has more computations needed. We can see that more steps require more computation in return for better generalization performance as evidenced in Table 2.

| Inner Loop Steps | 1 | 2 | 3 | 4 | 5 |
|---|---|---|---|---|---|
| MAML ++ (ms/iter) | 275.319 | 433.8172 | 579.314 | 786.3278 | 947.0376 |
| MAML (ms/iter) | 294.373 | 475.1218 | 658.4436 | 859.1158 | 1028.1656 |

Table 4: MAML++ Omniglot 5-way Results: *MAML++* indicates MAML + all the proposed fixes. We report our own base-lines on MAML to provide better relative intuition on how each method impacted the test accuracy of the model. We can see that MAML++matches or improves on MAML across all cases. MAML++also has performance very close to Meta-SGD which uses double the amount of parameters that MAML requires (about 40K extra parameters), whilst only using 1 extra parameter, per layer, per step for LSLR and $(f \times l \times (s - 1))$ (where $f$ is the number of filters at layers preceding batch normalization, $l$ is number of layers and $s$ is number of inner loop step) when using BNWB. Having less parameters also means smaller training and testing times.

| Omniglot 5-way Few-Shot Classification | | |
|---|---|---|
| | **Accuracy** | |
| **Omniglot** | **1-shot** | **5-shot** |
| Siamese Nets | 97.3% | 98.4% |
| Matching Nets | 98.1% | 98.9% |
| Neural Statistician | 98.1% | 99.5% |
| Memory Mod. | 98.4% | 99.6% |
| Meta-SGD | **99.53$\pm$0.26%** | **99.93$\pm$0.09%** |
| Meta-Networks | 98.95% | - |
| MAML (original) | 98.70$\pm$0.4% | 99.9$\pm$0.1% |
| MAML (local replication) | 98.57% | 99.82% |
| MAML++ | 99.47% | 99.93% |

