# OpenReview forum: "How to train your MAML"
_ICLR.cc/2019/Conference_

### Official Review · AnonReviewer1 · 2018-10-31
**Improving MAML**

**Rating:** 7
**Confidence:** 4

**Review:**

Paper summary - This paper provides a bag of sensible tricks for making MAML more stable, faster to learn, and better in final performance.
Quality - The quality of the work is strong: the results demonstrate that tweaks to MAML produce significant improvements in performance. However, I have some concern that certain portions of the text overclaim (see concerns section below).
Clarity - The paper is reasonably clear, with some exceptions (see concerns section).
Originality - The techniques described in the paper range from only mildly novel (e.g. MSL, DA), to very obvious (e.g. CA). Additionally, the paper's contributions amount to tweaks to a previously existing algorithm.
Significance - The quality of the results make this a significant contribution in my view.
Pros - Good results on a problem/algorithm of great current interest.
Cons - Only presents (in some cases obvious) tweaks to a previous algorithm; clarity and overclaiming issues in the writeup.

Concerns (please address in author response)
- The paper says  "we … propose multiple ways to automate most of the hyperparameter searching required". I'm not sure that this is true. The only technique that arguably removes a hyperparameter is LSLR. Even in this case, you still have to initialize the inner loop learning rates, so I'm not convinced that even this reduces hyperparameters. Perhaps I've missed something, please clarify.
- Section 4's paragraph on LSLR seems to say that you have a single alpha for each layer of the network. If this is right, then saying your method has a "per layer gradient direction" is very confusing. Each layer's alpha modulates the magnitude of that layer's update vector, but not its direction. The per-layer alphas together modify the direction of the global update vector. Perhaps I've misunderstood; equations describing exactly what LSLR does would be helpful. In any case, this should be clarified in the text.

Suggestions (less essential than the concerns above)
- The write-up is redundant and carries unnecessary content. The paper would be better shorter (8 pages is not a minimum :)
Section 1 covers a lot of background on the basics of meta-learning background that could be skipped. Other papers you cite (e.g. the MAML paper cover this).
    - Section 2 goes into more detail about e.g. matching nets than is necessary.
    - Section 2 explains MAML, which is then covered in much more detail in Section 3; better to leave out the Section 2 MAML paragraph.
    - Sections 3 and 4 are very redundant. Combine them for a shorter (i.e., better!) paper.
- The paper says, "Furthermore, for each learning rate learned, there will be N instances of that learning rate, one for each step to be taken. By doing this, the parameters are free to learn to decrease the learning rates at each step which may help alleviate overfitting." Does this happen empirically? Space could be freed up (see above) to have a figure showing whether or not this happens.
- The paper says, "we propose MAML++, an improved meta-learning framework" -- it's a little too far to call this a new framework. it's still MAML, with improvements.

Typos
- "4) increase the system’s computational overheads" -> overhead
- "composed by" -> composed of
- "Santurkar et al. (2018).", "Krizhevsky et al. (2012),",  "Finn et al. (2017) " -> misplaced citation parens
- "a method that reduce" -> reduces
- "An evaluation ran consisted" -> evaluation consisted
- The Loshchlikov and Hutter citation in the bibliography isn't right. It should be "Sgdr: Stochastic gradient descent with restarts." (2016) instead of "Fixing weight decay regularization in adam" (2017).

---

> ### Author Response · Authors · 2018-11-17
> **Response to Reviewer 1**
>
> Thanks for taking the time to review our paper. Further thanks for your very detailed, useful and constructive comments. We will now address your concerns below in the same order they were made:
>
> We claim that we reduce the hyperparameter choices needed because once our methodologies are applied exactly as proposed, the resulting system will achieve very high generalization and fast convergence without any additional tuning. We have attempted to initialize the learning rates from a random uniform distribution (ranging from 0.1 to 0.01) in addition to initializing manually. Both methods, interestingly, converge to very similar learning rates. Thus, random initialization suffices for that aspect, which does reduce the need for explicitly choosing a learning rate.
> Regarding the gradient directions. The alpha also includes a sign. So, in other words, the alpha also learns the direction of the learning rate, hence our claim. In fact, an interesting finding is that, in specific steps and layers, the network chooses to “unlearn” or flip the sign of the learning rate. Further investigation is required to understand this behavior, but a current working hypothesis is that the network is trying to “forget” particular parts of its weights, which somehow produces more efficient learning, in subsequent steps. We will further expand on this in a future blog post.
>
> All of your suggestions and typo-locations are spot-on and we will take care to address all of those in the final version of the paper. Again, we really thank you for providing such a detailed and constructive review.

---

> > ### Comment · AnonReviewer1 · 2018-11-23
> > **response to authors**
> >
> > < The alpha also includes a sign. >
> > Ok that makes sense. It might be worth adding a sentence that says this (if there isn't one already).
> >
> > < Thus, random initialization suffices for that aspect, which does reduce the need for explicitly choosing a learning rate. >
> > Ok, so you're saying that one of maml's hyperparameters is now a set of less-sensitive hyperparameters. That sounds useful, but it's very different from from the claim you make in the paper that maml++ gives "automatic learning for most of the system’s hyperparameters". There are two problems with this claim
> >
> > 1.) As far as I see, the only thing you've _automated_ is the setting of the inner loop learning rate, and in so doing you added more hyperparameters that need to be set. It's good they're not so sensitive, but they still have to be set. It's also good that your settings make the system overall easier to optimize, but that's not the same as automation.
> > 2.) You haven't gotten rid of "most" of the hyperparameters. There's still the outer loop learning rate and the other optimizer hyperparameters (e.g. \beta_1 and \beta_2 in Adam). In the most generous interpretation, you've made half of the hyperparameters less sensitive. Additionally, all of the architecture hyperparameters e.g. number of layers, number of units per layer, etc etc still need to be set by the user.
> >
> > Overall, this seems to me like a significant over-claiming issue. Replacing the language about "automating most hyperparameters" with something about "reducing inner loop hyperparameter sensitivity" would be sufficient.

---

> > > ### Author Response · Authors · 2018-11-23
> > > **2nd Response to Reviewer 1**
> > >
> > > Thanks for your prompt response. I think your point about the _automation_ of things is correct. I will amend the paper to be more precise in that claim as per your request. Regarding the automation of additional parts of the system, I am currently working on that, but it felt like it exceeded the scope of this paper hence breaking it into smaller easier to digest papers, that tackle one thing at a time. In my experience, papers that try to do too many things at once are often incredibly hard to write, and even harder to read.
> > >
> > > I will modify the particular claim shortly. Thanks for your time.

---

### Official Review · AnonReviewer3 · 2018-11-02
**In-depth discussions and improvements on MAML**

**Rating:** 6
**Confidence:** 5

**Review:**

In the work, the authors improve a simple yet effective meta-learning algorithm called Model Agnostic meta-learning (MAML) from various aspects including training instability, batch normalization etc. The authors firstly point out the issues in MAML training and tackle each of the issue with a practical alternative approach respectfully. The few-shot classification results show convincing evidence.

Some major concerns:
1. The paper is too specific about improving one algorithm, the scope of the research is quite narrow and I'm afraid that some of the observations and proposed solutions might not generalize into other algorithms;
2. Section 4, "Gradient Instability → Multi-Step Loss Optimization." I don't see clearly why the multi-step loss would lead to stable gradients. It causes much more gradient paths than the original version. I do see the point of weighting the losses from different step;
3. The authors should have conducted careful ablation study of each of the issues and solutions. The six ways of proposed improvements may make the the performance boost hard to understand. It would help to see which way of the proposed improvement contribute more than others;
4. Many of the proposed improvements are essentially utilizing annealing mechanisms to stabilize the training, including 1) anneals the weighting of the losses from different step; 2) anneal the second derivative  to the first derivative;
5. For the last two improvements about the learning rate, there are dozens of literature on meta-learning learning rate and the proposed approach does not seem to be novel;

Minors
1. The reference style is inconsistent across the paper, sometimes it feels quite messy. For example, "Batch Stochastic Gradient Descent Krizhevsky et al. (2012)" "Another notable advancement was the gradient-conditional meta-learner LSTM Ravi & Larochelle (2016)";
2. Equation (2) (3) the index b should start from 1, size of B should be 1 to B;

---

> ### Author Response · Authors · 2018-11-17
> **Response to Reviewer 3**
>
> Thank you for taking the time to review our paper. Before I start delving into the technical aspects of this response. To address your concerns, I will use an enumeration that matches the indexes of your concerns.
> The paper is indeed targeted towards a particular class of algorithms. That class being end-to-end differentiable gradient-based meta-learning. MAML and Meta-learner LSTM [1] are two instances of that particular class of algorithms. Our proposed techniques can be applied to any algorithm of that class, given that they utilize inner-loop optimization processes as part of their learning. So, even though this work is indeed targeted towards a particular class of models, that class is general enough and applicable to enough domains that we felt that an investigation of the type presented in this paper was necessary. In fact, the work in this paper was the result of the first author’s attempts to build systems that learn various other components (i.e. instead of just learning a highly adaptable parameter initialization, he was attempting to learn loss functions/update functions and dynamic generation of parameter initializations given a task among others). What he realized, however, was that MAML was really hard to actually work with, being very inflexible to architecture configuration, causing gradient degradation problems, instability in training and requiring lots of manual inner loop learning rate tuning.  In attempting to fix those problems, so he could build on top more complicated systems, this paper came to be.
> In MAML, the resulting inference model is effectively an unrolled 5 layer network over N steps. If that N=5, then the resulting model has a depth of effectively 25 layers. In standard deep networks, gradient degradation can be greatly reduced or altogether removed via the usage of skip-connections. Since in MAML we can’t really apply skip-connections from a subsequent model to a previous one (because that would further complicate the gradients), we decided that the best way to inject clean/stable gradients to all iterations of the network would be to use 2 losses for each step-wise network. One loss, providing an implicit gradient, coming from subsequent iterations of the network (i.e. the original MAML loss), and another per-step loss, providing an explicit gradient, coming directly from evaluating the model on the target set. This way, every network iteration receives stable gradients which keep the network stable during the early epoch training. Eventually, the importance of earlier steps becomes 0, which means that the original MAML loss is used instead. However, since the network has already learned a stable parameterization, the stability remains throughout training (we empirically confirmed this).
> We conducted an ablation study on 20-way 1-shot Omniglot, as shown in table 2. We did want to conduct even more exhaustive ablation studies across all Omniglot and Mini-Imagenet tasks, however, due to computing constraints we had to restrict ourselves. Using the “hardest” Omniglot 20-way 1-shot task as the ablation study’s subject seemed like a sensible thing to do since it was cheaper computationally, but “hard” enough for the results to generalize well in other tasks.
> Indeed, annealing various components is not as novel as some of the other proposals in the paper. However, since this paper was essentially an engineer’s handbook on how to train MAML-like models, we felt that people should be aware of the effect those techniques have on the system’s performance.
> Indeed, there is other literature on meta-learning learning rates. Our approach’s novelty lies in learning “per-step” and “per-layer” learning rates. By being able to learn per step learning rates, we allow the network to choose to decrease or increase it’s learning rates at each step, to minimize overfitting. Another interesting phenomenon, that we will address in a future blog post, is the fact that across all networks, we noticed that particular layers choose to “un-learn” (flipping the direction of the learning rate) at particular steps. We theorize that the network might be attempting to remove some existing knowledge to replace it with new knowledge, or using forgetting as a way to steer gradients for more efficient learning.
>
> Regarding the minor concerns, yes, we will fix the referencing inconsistencies and the batch size indexing problem.
>
> Once again, I want to thank you for taking the time to review our work.
>
> 1. Ravi, S. and Larochelle, H. (2016). Optimization as a model for few-shot learning.

---

### Official Review · AnonReviewer2 · 2018-11-02
**A paper with marginal novelty over an established framework.**

**Rating:** 5
**Confidence:** 3

**Review:**

[Summary]
This work presents several enhancements to the established Model-Agnostic Meta-Learning (MAML) framework. Specifically, the paper starts by analyzing the issues in the original implementations of MAML, including instability during training, costly second order derivatives evaluation, missing/shared batch normalization statistics accumulation/bias, and learning rate setting, which causes unstable or slow convergence, and weak generalization. The paper then proposes solutions corresponding to each of these issues, and reports improved performance on benchmark datasets.

Pros
Good technical enhancements that fix some issues of a popular meta-learning framework
Cons
Little conceptual and technical novelty

[Originality]
The major problem I found in this work is the lack of conceptual and technical novelty. The paper basically picks up some issues of the well-established MAML framework, and applies some common practices or off-the-shelf technical treatments to fix these drawbacks and improve the training stability, convergence, or generalization, etc. E.g., it seems to me that the most effective enhancement comes from the use of adoption of learning rate setting (LSLR), or variant version of batch normalization (BNWB+BNRS) in Table 1, which have been the standard tricks to improve performance in the deep learning literature. Overall, the conceptual originality is little.

[Quality]
The paper does get most things well executed from the technical point of view. There does not seem any major errors to me. The results reported are also reasonable within the meta-learning context, despite lack of originality.

[Clarity]
The paper is generally well written and I did not have much difficulty to follow.

[Significance]
The significance of this work is marginal, given the lack of originality. The technical enhancements presented in the paper, however, may be of interest to people working in this area.

---

> ### Author Response · Authors · 2018-11-17
> **Response to Reviewer 2**
>
> Thank you for your review.
>
> Regarding the conceptual and technical novelty concerns.
>
> To clarify, our main contribution comes in the form of carrying an investigation on how MAML can be stabilized and how the model can be modified such that it can consistently achieve faster convergence and strong generalization results without any hyperparameter tuning required. Then, once the investigation is completed and key problem-areas isolated, we use our investigation insights to improve the system. In fact, the whole reason for doing this was because we attempted to built new research ideas on top of MAML only to find out just how sensitive and unstable the system was. Therefore, we decided that finding the issues and fixing them would enable researchers working on gradient-based end-to-end meta-learning, such as MAML or Meta Learner LSTM [1] to concentrate on the new approach they want to build rather than trying to overcome instability issues of the base methodology. Furthermore, the industry would also benefit from this, as they would have an easier time training MAML based models.
>
> Most of the proposed approaches are novel and non-obvious (i.e. LSLR, BNWB+BNRS, and multi-step loss optimization). Overcoming gradient degradation issues by utilizing multi-step target-loss optimization which is annealed over time, is in our knowledge, done for the first time in this work. Furthermore, we provide novel contributions in the form of learning things “step-by-step”.
>
> For example, we propose that learning per-layer, per-step learning rates would benefit the system, more so than just learning per-layer learning rates and sharing them. The reason is that the model would be free to choose to decrease its learning rate or otherwise change it from step to step to reduce overfitting. This technique is both novel and non-obvious. Furthermore, LSLR is not something that is possible in standard deep learning, as learning the learning rates would require an additional level of abstraction (thus entering the meta-learning arena).
>
> Another contribution with significant novelty comes in the form of proposing a step-by-step batch norm variant, designed for meta-learning systems that require inner loop optimization. Learning batch norm parameters for every step, as well as collecting per-step running statistics speeds up the system and allows batch normalization to truly work in this setting, whereas the previous variant of batch norm used, constrained things further, instead of achieving the improved convergence and generalization that batch norm can achieve in standard deep learning training setups.
>
> The rest of the contributions, such as annealing the derivative order and using cosine scheduling for Adam are less novel, but nonetheless important to investigate. We show from our experiments that those approaches can improve the system, something which was previously unconfirmed.
>
> The comparative performance (between MAML and MAML++) both in convergence speed and final generalization is significant and produces state of the art results. Furthermore, that performance is achieved far more consistently and with more stability across architectures. We hold the belief that the community would really benefit from this work, hence why we submitted it.
>
> 1. Ravi, S. and Larochelle, H. (2016). Optimization as a model for few-shot learning.

---

### Public Comment · (anonymous) · 2018-09-30
**Related works that have better results are missing?**

https://arxiv.org/pdf/1805.08311.pdf has better Omniglot 5-way results and better Mini-Imagenet 5-way results
https://arxiv.org/pdf/1707.03141.pdf has better Mini-Imagenet 5-way results
https://arxiv.org/pdf/1807.02872.pdf has better Mini-Imagenet 5-way 5-shot results

---

> ### Author Response · Authors · 2018-09-30
> **Re: Related Works with better results**
>
> Thanks for your comment. Firstly, I'll reiterate that the main point of the paper is to improve MAML as a model itself. Furthermore, we did a very thorough literature review but missed out on the papers you have stated. The work in our paper had already taken full shape in May thus meaning that works 1 and 3 (that came later) escaped our radar. The second paper you mentioned, "Neural Attentive Meta Learner" was not included in many of the latest few-shot learning papers that came out in June 2018, thus making it harder for us to be aware of it. We did try to cover everything in the literature prior to starting our work, however as is often the case, one or two papers might escape ones review. Especially in this field, where papers keep coming out on a daily basis on arxiv. We shall add the approaches you mentioned in our result tables when editing is allowed again. Thank you for informing us of some literature we were previously unaware of.

---

### Public Comment · (anonymous) · 2018-10-02
**Method of Learning Gradient Directions Not Clear**

When describing LSLR you likened your method to Meta-SGD, but in Meta-SGD the gradient direction is represented by the optimizer parameters \alpha which has the same dimensionality as the learner parameters \theta. In your method you claim that you reduce computational costs by learning "per layer per step" learning rates and directions. Can you please clarify how are your directions represented if not with the same number of parameters as used in Meta-SGD?

---

> ### Author Response · Authors · 2018-10-02
> **Method of Learning Gradient Directions Clarification**
>
> Meta-SGD learns alphas of dimensionality equal to the network parameters. Instead with LSLR we propose learning one alpha for each layer of the network. A component qualifies as a layer if it has learnable weights or biases in it. In addition, instead of just learning a learning rate and direction (alpha) for each layer to be used across all inner loop steps, we instead propose to learn different alphas for each inner loop step. This allows the network to choose to decay its alphas or otherwise change them, to maximize generalization performance (in some cases we noticed the network choosing to unlearn for some inner loop steps by using a negative learning rate and learn in others). So, to summarise, we learn one learning rate and direction for each layer for any given inner loop step. The network we used had 4 CNN layers along with a final softmax. That's a total of 5 layers, but since we learn learning rates for weights and biases separately, this means that the model learns a total of 10 learning rates and directions for any given step. For example, in the case where the model takes 5 inner loop steps, we have a total of 5 x 10 = 50 learning rates and directions, which is represented by 50 learnable parameters in the system.

---

### Meta-Review · Area_Chair1 · 2018-12-14

**Confidence:** 4
**Recommendation:** Accept (Poster)

**Metareview:**

This paper proposes several improvements for the MAML algorithm that improve its stability and performance.
Strengths: The improvements are useful for future researchers building upon the MAML algorithm. The results demonstrate a significant improvement over MAML. The authors revised the paper to address concerns about overstatements
Weaknesses: The paper does not present a major conceptual advance. It would also be very helpful to present a more careful ablation study of the six individual techniques.
Overall, the significance of the results outweights the weaknesses. However, the authors are strongly encouraged to perform and include a more detailed ablation study in the final paper. I recommend accept.